# Quantitative SARS-CoV-2 subgenomic RNA as a surrogate marker for viral infectivity: Comparison between culture isolation and direct sgRNA quantification

Rossana Scutari[1,2☯], Silvia Renica[1☯], Valeria Cento[3,4], Alice Nava[5], Josè Camilla Sammartino[6], Alessandro Ferrari[6], Arianna Pani[5], Marco Merli[7], Diana Fanti[5], Chiara Vismara[5], Francesco Scaglione[1,5], Massimo Puoti[7], Alessandra Bandera[8,9], Andrea Gori[8,9], Antonio Piralla[6], Fausto Baldanti[6,10], Carlo Federico Perno[2], Claudia Alteri[1,2]*

1 Department of Oncology and Hemato-oncology, University of Milan, Milan, Italy, 2 Multimodal Research Area, Bambino Gesù Children Hospital IRCCS, Rome, Italy, 3 Department of Biomedical Sciences, Humanitas University, Pieve Emanuele, Milan, Italy, 4 IRCSS Humanitas Research Hospital, Rozzano, Milan, Italy, 5 Chemical-Clinical and Microbiological Analysis, ASST Grande Ospedale Metropolitano Niguarda, Milan, Italy, 6 Microbiology and Virology Department, Fondazione IRCCS Policlinico San Matteo, Pavia, Italy, 7 Infectious Diseases Unit, Azienda Socio-Sanitaria Territoriale (ASST) Grande Ospedale Metropolitano Niguarda, Milan, Italy, 8 Infectious Diseases Unit, Fondazione IRCCS Ca' Granda Ospedale Maggiore Policlinico, University of Milan, Milan, Italy, 9 Department of Pathophysiology and Transplantation, University of Milan, Milan, Italy, 10 Department of Clinical, Surgical, Diagnostic and Pediatric Sciences, University of Pavia, Pavia, Italy

☯ These authors contributed equally to this work.
* claudia.alteri@unimi.it

**Data Availability Statement:** All relevant data are within the paper and its Supporting Information files.

## Abstract

Detection of subgenomic (sg) SARS-CoV-2 RNAs are frequently used as a correlate of viral infectiousness, but few data about correlation between sg load and viable virus are available. Here, we defined concordance between culture isolation and E and N sgRNA quantification by ddPCR assays in 51 nasopharyngeal swabs collected from SARS-CoV-2 positive hospitalized patients. Among the 51 samples, 14 were SARS-CoV-2 culture-positive and 37 were negative. According to culture results, the sensitivity and specificity of E and N sgRNA assays were 100% and 100%, and 84% and 86%, respectively. ROC analysis showed that the best E and N cut-offs to predict positive culture isolation were 32 and 161 copies/mL respectively, with an AUC (95% CI) of 0.96 (0.91–1.00) and 0.96 (0.92–1.00), and a diagnostic accuracy of 88% and 92%, respectively. Even if no significant correlations were observed between sgRNA amount and clinical presentation, a higher number of moderate/severe cases and lower number of days from symptoms onset characterized patients with sgRNA equal to or higher than sgRNA cut-offs. Overall, this study suggests that SARS-CoV-2 sgRNA quantification could be helpful to estimate the replicative activity of SARS-CoV-2 and can represent a valid surrogate marker to efficiently recognize patients with active infection. The inclusion of this assay in available SARS-CoV-2 diagnostics procedure might help in optimizing fragile patients monitoring and management.

**Funding:** this work is supported by STOP-COVID project, founded by Fondazione IRCCS Ca' Granda Ospedale Maggiore Policlinico. The funder had no role in study design, data collection, analysis, decision to publish, or preparation of the manuscript. The recipient of the fund is the corresponding author, Claudia Alteri.

**Competing interests:** The authors have declared that no competing interests exist.

## Introduction

In early 2020, SARS-CoV-2 appeared and quickly spread globally [1]. In order to promptly identify positive cases, its detection is rapidly committed to qualitative real-time reverse transcription PCRs (q-rtPCR), considered today the gold standard for SARS-CoV-2 diagnosis [2]. However, these methods are not designed to provide a quantitative SARS-CoV-2 RNA and cannot distinguish between replicating virus and residual genomic material [3, 4]. To date, the best indicator of replicating virus is SARS-CoV-2 culture isolation. However, this technique requires a biosafety level 3, highly specialized personnel, high costs, it is time-consuming and needs of at least 7 days to provide a certain negative result.

Due to limitations of q-rt-PCR and virus isolation methods, the first unable to assess active viral replication and the last characterized by long processing times, there is a strong need for a simple and rapid test that can provide accurate and rapid results on SARS-CoV-2 residual replication capacity, especially from the perspective of long-term positivity to SARS-CoV-2 and clinical management of SARS-CoV-2 positive patients.

Previous studies suggested that, in clinically resolved patients, the presence of persistently detectable SARS-CoV-2 by q-rt-PCR over two or more weeks might be related to the elimination of residual viral genomic material rather than to the actual replicative potential of the virus itself [5, 6].

Of note, SARS-CoV-2 has a complex replication cycle characterised by a discontinuous transcription process, resulting in subgenomic RNAs (sgRNAs) production [7]. These sgRNAs are susceptible to enzymatic degradation and are hardly present in virions [8]. Since the process of sgRNA formation only occurs during genomic replication and transcription, several studies suggested that sgRNA can be used as a correlate of viral infectiousness [9–16].

However, few data are available so far regarding the role of quantitative determination of these sgRNA and potential correlation with the SARS-CoV-2 infectivity [17].

## Material and methods

### Study population

The study was conducted between October 2020 and November 2021 at ASST Grande Ospedale Metropolitano Niguarda, Milano and Fondazione IRCCS Policlinico San Matteo, Pavia. During this period, a total of 110 nasopharyngeal swabs (UTM$^{TM}$, Copan Italia, Brescia, Italy) (1 sample per patient) were tested for SARS-CoV-2 infectivity by culture isolation for diagnostic purposes. Swabs collected to a maximum of 14 days after symptoms onset (usually defined as the temporal window for the persistence of viral shedding in upper respiratory tract [18, 19] (n = 51, S1 Fig) were retrospectively selected to compare the culture results with SARS-CoV-2 genomic and sgRNA quantifications.

For each patient, demographic and clinical information such as age, gender, clinical manifestations and symptoms were retrieved and stored in an anonymous database ad hoc built for the study.

The severity of COVID-19 was classified, in according with WHO scale [20], into asymptomatic, mild and moderate/severe.

### Ethical committee

The study protocol was approved by local Research Ethics Committee of the Niguarda and San Matteo hospitals (prot. 92–15032020 and P_20200029440). This study was conducted in accordance with the principles of the 1964 Declaration of Helsinki. Informed consent was waived in accordance with the regulations on observational retrospective studies.

### SARS-CoV-2 load and subgenomic RNA quantification by ddPCR

SARS-CoV-2 genomic RNA and sgRNA were quantified by means of the QX200™ Droplet Digital™ PCR System (ddPCR, Bio-Rad Laboratories, Inc.). In detail, one home-made and two previously tested assays [21] targeting 3 different regions of RNA-dependent RNA-polymerase (RdRp) of SARS-CoV-2 were used to quantify SARS-CoV-2 genomic RNA. The assay targeting the RNAseP housekeeping gene was used as reference [22]. The sgRNAs were quantified using assays adapted for the ddPCR system and targeting the envelope and nucleocapsid transcripts [10, 23]. Primer and Probe used to detect and quantify genomic and subgenomic RNA are reported in S1 Table. SARS-CoV-2 viral load and sgRNA were expressed in number copies/mL of swab. Full protocol used to detect and quantify SARS-CoV-2 sgRNAs and viral load was reported in S1 File.

### Culture isolation

For virus isolation, all samples were inoculated into a Vero E6 (VERO C1008 -Vero 76, clone E6, Vero E6; ATCC1 CRL-1586TM) confluent 24-well microplate between 8 and 24 h after positivity results. After 1 h incubation at 33˚C in 5% $CO_2$ in air, the inoculum was discarded and 1 mL of medium (Eagle's minimum essential medium supplemented with 1% penicillin, streptomycin, and glutamine and trypsin at 5 mg/mL) was added to each well. Cells were incubated at 33˚C in 5% $CO_2$ in air. After incubation for 7 days, 200 µl of supernatant from a well showing a cytopathic effect was tested for the presence of SARS-CoV-2 by molecular assay (gene E real-time RT-PCR) [24].

### Statistics and reproducibility

Descriptive statistics are expressed as median values and interquartile range (IQR) for continuous data and number (percentage) for categorical data. To assess significant differences Fisher exact and Mann–Whitney or Kruskal-Wallis tests were used for categorical and continuous variables, respectively. All quantifications were performed in duplicate.

To define the performance of in-house ddPCR assays, sensitivity, specificity, positive and negative predictive values (PPV and NPV) were assessed against the culture isolation, considered as the gold standard for SARS-CoV-2 active virus replication. A receiver operator characteristic (ROC) curve was performed to determine the optimal cut-off point to identify true-positive. Potential association between SARS-CoV-2 sgRNA and genomic RNA values and clinical presentation were also evaluated.

Statistical analyses were performed with SPSS software package for Windows (version 23.0, SPSS Inc., Chicago, IL) and Rgui (v.4.2.3). Figures were generated by GraphPad Prism 8. A p-value <0.05 was considered statistically significant.

## Results

### Patient's characteristics

The demographic and clinical characteristics of 51 patients included in the study are reported in Table 1. Twenty-five patients were male (59.0%) with a median age of 59 (Interquartile range [IQR]: 41–69) years.

At first SARS-CoV-2 positivity, clinical information related to COVID-19 was retrieved for 17 individuals. Among them, 7 (41.2%) were asymptomatic, 5 (29.4%) with moderate/severe infection and 5 (29.4%) with mild infection. Four patients reported fever (23.5%) and 5 (29.4%) had evidence of interstitial pneumonia. Immunocompromised status was observed in 6 (35.3%) patients. Median days from symptoms onset were 8 (IQR: 1–11).

**Table 1. Demographic and clinical characteristics of the study population.**

| | Overall | Culture isolation | |
| --- | --- | --- | --- |
| | | **SARS-CoV-2 Positive culture isolation** | **SARS-CoV-2 Negative culture isolation** |
| **Patients, N** | 51 | 14 | 37 |
| **Males** | 25 (49.0) | 9 (64.3) | 16 (43.2) |
| **Age (years)** | 59 (41–69) | 60 (44–67) | 58 (38–71) |
| **SARS-CoV-2 PCR positivity[a]:** | | | |
| one gene target | 24 (64.9) | 12 (92.3) | 12 (50.0) |
| >one gene target | 13 (35.1) | 1 (7.7) | 12 (50.0) |
| **Days from symptoms onset** | 8 (1–11) | 2 (1–7) | 10 (3–12) |
| **COVID-19 manifestation at first positive nasopharyngeal swab[b]** | | | |
| Asymptomatic | 7 (41.2) | 1 (25.0) | 6 (46.2) |
| Mild | 5 (29.4) | 1 (25.0) | 4 (30.7) |
| Moderate/severe | 5 (29.4) | 2 (50.0) | 3 (23.1) |
| **Specific symptoms at first nasopharyngeal swab[b]** | | | |
| Fever | 4 (23.5) | 0 (0.0) | 4 (30.8) |
| Cough | 1 (5.9) | 1 (25.0) | 0 (0.0) |
| Dyspnea | 2 (11.8) | 0 (0.0) | 2 (15.4) |
| Evidence of interstitial pneumonia | 5 (29.4) | 2 (50.0) | 3 (23.1) |
| **Immunocompromised patients[b]** | 6 (35.3) | 3 (75.0) | 3 (23.1) |

Data are expressed as median (interquartile range, IQR), or N (%). COVID-19, Coronavirus Disease 2019

[a]Available for 37 patients

[b]Available for 17 patients.

Regarding SARS-CoV-2 isolation by cell culture, 14 (27.5%) nasopharyngeal swabs resulted to be SARS-CoV-2 positive at culture isolation, while the remaining 37 (72.5%) resulted to be SARS-CoV-2 negative.

## SARS-CoV-2 subgenomic results against culture

E and N sgRNAs were detected in 20/51 (39.2%) and 19/51 (37.3%) nasopharyngeal swabs, respectively. The median (IQR) SgRNA was 1,295 (238–906,850) and 1,680 (325–254,240) copies/mL, respectively (S2 Table).

Stratifying the population according to culture result, E SgRNA was detected in all 14 culture positive swabs and only in 6/37 culture negative swabs (p-value<0.001) with a median (IQR) load of 140,700 (350–1,071,000) and 406 (210–770) copies/mL, respectively (Fig 1 Panel A and S2 Table). N sgRNA was detected in 14/14 culture positive swabs and only in 5/37 culture negative swabs (p-value<0.001), with a median (IQR) load of 34,069 (325–306,600) and 770 (630–980) copies/mL, respectively (Fig 1 Panel B and S2 Table).

Direct quantification of genomic SARS-CoV-2 load revealed the presence of SARS-CoV-2 RNA in 37/51 samples, with a median (IQR) load of 5,899 (705–105,817) copies/mL. In line with the sgRNA results, SARS-CoV-2 load was detected in all 14 culture positive swabs and in 23/51 culture negative swabs (p-value = 0.005), with a higher viral load in culture positive respect to culture negative swabs (median, IQR: 3,174,612 [46,650–10,000,000] vs 1,913 [411–8,680] copies/mL, p-value<0.001) (Fig 1 Panel C and S2 Table).

According to culture isolation results, the sensitivity and specificity observed for the sgRNA E assay were 100% (95% CI: 77%-100%) and 84% (95%CI: 68%-94%), respectively with a PPV value of 70% (95%CI: 53%-83%) and NPV value of 100% (95%CI: 89%-100%). Similarly, the SgRNA N assay showed a sensitivity of 100% (95%CI: 77%-100%), a specificity of 86% (95%CI:

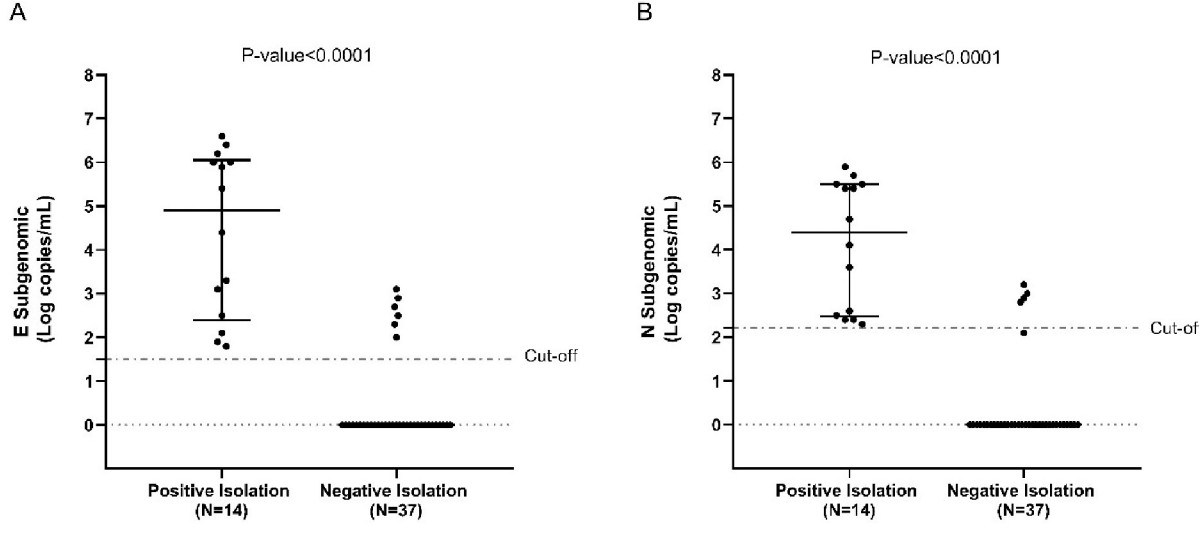

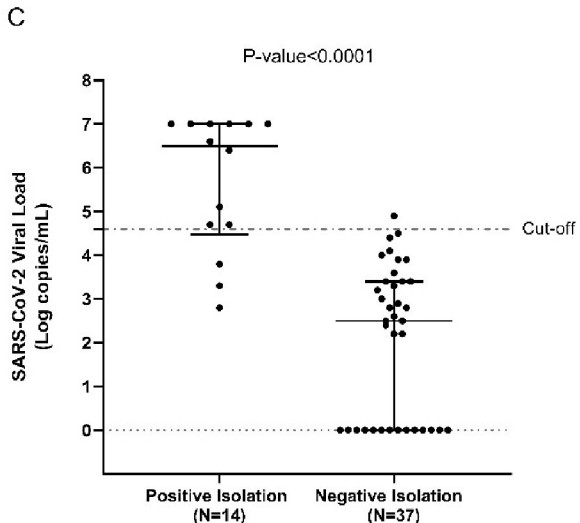

**Fig 1. SARS-CoV-2 subgenomic and genomic load against culture results.** The dot plots represent the quantification of **(A)** E subgenomic, **(B)** N subgenomic, and **(C)** SARS-CoV-2 RNA load in positive and negative cultures. The dotted line represents the best cut-off, calculated by ROC analysis, to predict a positive culture isolation. The bars indicate the median and interquartile range values (IQR).

71%-95%), and PPV and NPV values of 74% (95%CI: 55%-86%) and 100% (95%CI: 89%-100%), respectively. The diagnostic accuracy was 88% (95%CI: 76%-96%) for sgRNA E and 90% (95%CI: 79%-97%) for sgRNA N.

As expected and due to the renowned ddPCR high sensitivity, genomic SARS-CoV-2 assay showed a sensitivity and specificity against culture of 100% (95%CI: 77%-100%) and 38% (95% CI: 22%-55%), respectively, with PPV value of 38% (95%CI: 32%-44%) and NPV value of 100% (95%CI: 77%-100%).

Concomitant RNAseP RNA quantification confirmed the high quality of all the 51 naso-pharyngeal swabs (S2 Fig).

## SARS-CoV-2 culture isolation vs SARS-CoV-2 SgRNA

By comparing culture isolation results and SgRNA detection, three categories were highlighted: negative concordant (negative culture isolation and negative sgRNA detection), positive concordant (positive culture isolation and positive sgRNA detection) and discordant (negative culture isolation and positive sgRNA detection). The characteristics of the population stratified against these categories is shown in S3 Table.

Out of the 37 negative culture swabs, 31 were confirmed negative to sgRNA detection (Concordant -/-), while 6 were tested positive (Discordant -/+). The 14 positive culture swabs were confirmed positive to sgRNA detection (Concordant +/+) (S3 Table). Of note, a clear trend of higher sgRNA values was found for +/+ samples respect to -/+ samples (sgRNA E, median [IQR] copies/mL: 140,700 [350–1,071,000] vs and 406 [210–770], p-value = 0.083; SgRNA N, median [IQR] copies/mL: 34,069 [325–306,600] vs 770 [630–980] copies/mL, p-value = 0.165) (Fig 2, Panel A and B).

Most of the 31 concordant -/- samples had also a quite lower SARS-CoV-2 viral load compared to the discordant -/+ and concordant (+/+) samples (median [IQR] copies/mL: 161 [0–910] copies/mL vs. 3,174,612 [46,650–10,000,000] vs. 7,572 [2,543–13,650], p-value<0.001) (Fig 2, Panel C).

## SARS-CoV-2 sgRNA load cut-off definition for viable virus

In order to define the best cut-off of sgRNA load to predict viable virus a ROC analysis was performed. Interestingly, ROC curve analysis identified 32 copies/mL for sgRNA E and 161 copies/mL for sgRNA N as the best cut-off to predict a positive culture isolation, with an AUC (95% CI) of 0.96 (0.91–1.00) for E sgRNA and 0.96 (0.92–1.00) for N sgRNA.

Regarding SARS-CoV-2 genomic load, the best cut-off to predict viable virus was 39,752 copies/mL, with an AUC (95% CI) of 0.93 (0.86–1.00).

By using the cut-off obtained with the ROC curve, the sensitivity and specificity against culture observed for the SgRNA E assay were 100% (95%CI: 77%-100%) and 84% (95%CI: 68%-94%), respectively with a PPV value of 70% (95%CI: 53%-83%) and NPV value of 100% (95% CI: 89%-100%). Similarly, the SgRNA N assay showed a sensitivity of 100% (95%CI: 77%-100%), a specificity of 90% (95%CI: 75%-97%), and PPV and NPV values of 78% (58%-90%) and 100% (95%CI: 89%-100%), respectively. The diagnostic accuracy shown by the assays applying the ROC cut-off was 88% (95%CI: 76%-96%) for sgRNA E and 92% (95%CI: 81%-98%) for sgRNA N.

Using the cut-off of 39,752 copies/mL, the genomic SARS-CoV-2 assay showed a sensitivity and specificity of 79% (95%CI: 49%-95%) and 97% (95%CI: 86%-100%), respectively, with PPV value of 92% (95%CI: 62%-100%) and NPV value of 92% (95%CI: 79%-98%) against culture.

## Correlation with clinical presentation

By correlating SARS-CoV-2 sgRNA and genomic RNA values with available clinical information, 2 out 4 (50.0%) patients with E and N sgRNA values equal to or higher than 32 and 161 copies/mL, respectively, were characterized by moderate/severe COVID-19 manifestations respect to the 3/13 (23.1%) patients with E and N sgRNA values below the cut-offs. Median (IQR) days from the onset of symptoms to the nasopharyngeal swab were quite lower in patients with E and N sgRNA values equal to or higher than the cut-offs respect to patients with E and N sgRNA values below the cut-offs (days: 5 [1–11] vs. 10 [4–15]). Superimposable results were found for patients with SARS-CoV-2 RNA values equal to or higher than 39,752 copies/mL respect to patients with SARS-CoV-2 RNA values below the cut-off.

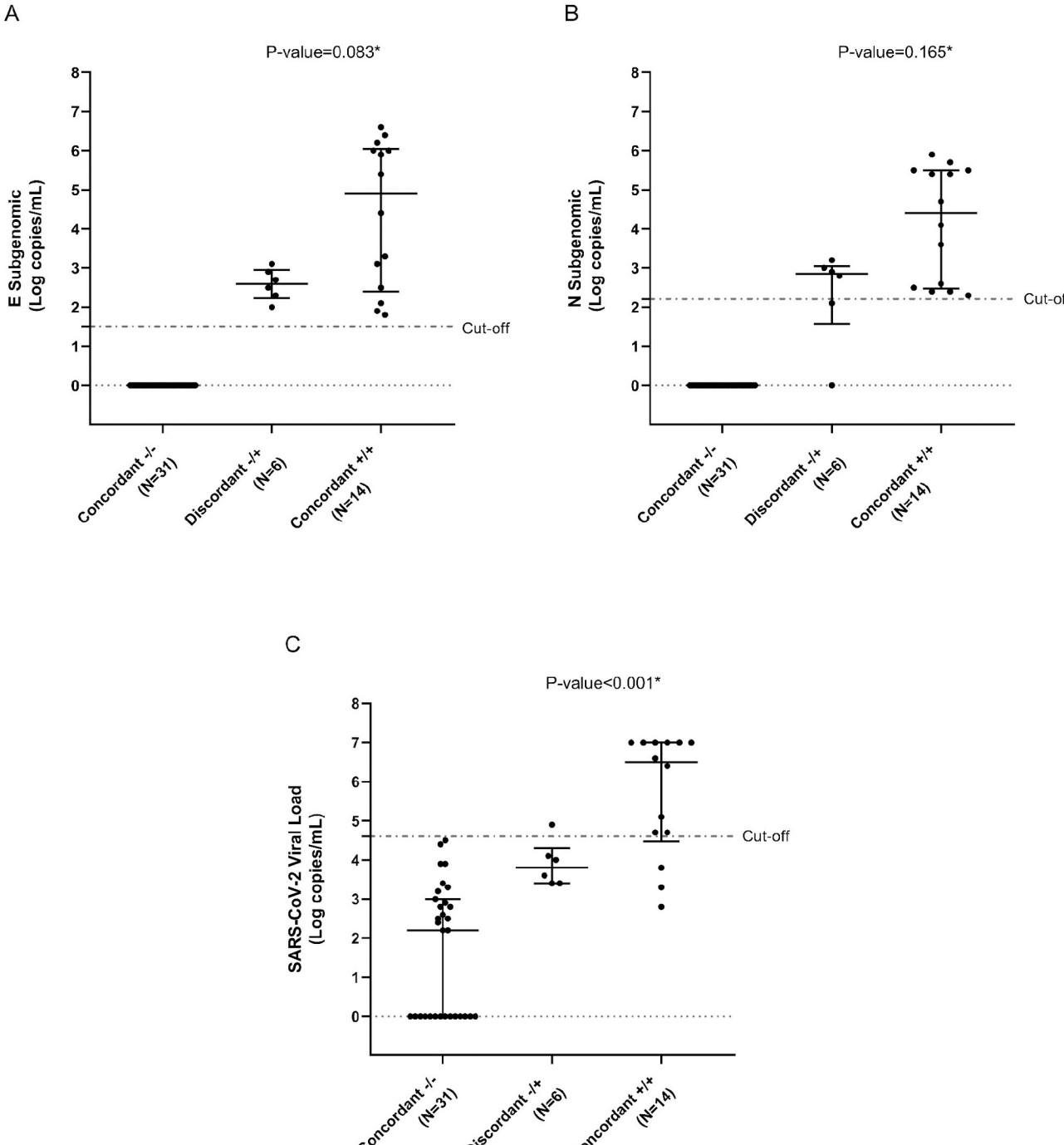

**Fig 2. SARS-CoV-2 subgenomic and genomic load against concordance between SARS-CoV-2 culture isolation and subgenomic RNA detection.**
The dot plots represent the quantification of (**A**) E subgenomic, (**B**) N subgenomic, (**C**) SARS-CoV-2 RNA load. Negative concordant is defined as negative culture isolation and negative sgRNA detection. Positive concordant is defined as positive culture isolation and positive sgRNA detection. Discordant is defined as negative culture isolation and positive sgRNA detection. SgRNA was considered positive (+) when at least one subgenomic RNA (N or E) was detectable. The dotted line represents the best cut-off, calculated by ROC curve analysis, to predict a positive culture isolation. *One-sided p-values comparing patients with concordant +/+ results and patients with discordant -/+ results were calculated by the Mann–Whitney test.

When clinical parameters were compared with SARS-CoV-2 culture isolation vs subgenomic RNA concordance, similar results were obtained (S3 Table).

## Discussion

This study demonstrates that SARS-CoV-2 sgRNA quantification might help to distinguish active viral replication from residual viral genetic material, thus suggesting that subgenomic SARS-CoV-2 detection and quantification could be used as a correlate of infectious viral shedding [15, 16]. In particular, we found that E and N sgRNA were always characterized by high load in positive SARS-CoV-2 culture samples, while were rarely detectable (with low quantification values) in culture-negative samples.

A number of studies recognized the role of sgRNA as an indicator of active virus replication and as a promising tool for patients' management [6, 10, 15, 16, 25], while others suggested that the detection of sgRNAs cannot represent a more useful marker in determining viability than genomic RNA, due to the similar decay of these two parameters [26, 27]. In some studies, specific sgRNAs can be detected at a very prolonged time from the onset of infection, even in non-immunocompromised individuals [28, 29], probably because tightly associated with membrane structures and thus protected from cellular RNases [30].

Our study aims to provide evidence that the quantification of sgRNAs, rather than their detection, can predict viable virus. In our study 32 copies/mL of sgRNA E and 161 copies/mL of sgRNA N were the best cut-off to predict culture isolation results, thus providing further confidence in applying direct sgRNA quantification assays to define risk of viable virus.

Regarding SARS-CoV-2 genomic load, the cut-off equal to 39,752 copies/mL showed a good specificity (97%) but a lower sensitivity than that observed for sgRNAs (79% vs. 100% of E and N sgRNA), thus confirming that the quantification of genomic RNA alone cannot adequately discriminate residual material from active replication.

In an exploratory analysis correlating sgRNA cut-offs with time from symptoms onset and disease manifestations, we observed that samples with sgRNA values equal or higher than the cut-offs estimated by ROC analyses had a sample date closer to symptoms onset and were characterized by more moderate/severe infections than samples with sgRNA values lower that the estimated cut-offs. Thus, quantification of sgRNA could be a useful surrogate for predicting not only active viral replication but also the duration of infectious viral shedding, as suggested by Perera et al. [6]. Worth of mention is that in fragile conditions like immunosuppression, duration of infectious viral shedding could be not only limited to the first days after the infection [31–33]. In line with this, in the subgroup of 59 nasopharyngeal swabs collected to more than 14 days from the symptoms onset and thus excluded by this analysis, four were positive for SARS-CoV-2 culture isolation and sgRNA quantification (median [IQR]: 11,320 [205–131,925] for N and 4,760 [1,917–15,873] for E). Three of these swabs were collected at 20-, 51- and 85-days post symptoms onset and belonged to immunocompromised individuals. Overall, these initial findings could suggest that application of these rapid sgRNA quantifications in clinical practice could help also in monitoring critically ill patients at high risk of severe manifestations.

Our study has some limitations. First, the sample size was small especially if considering samples with SARS-CoV-2 positive culture isolation. This limitation makes the results of the study exploratory. Therefore, more data on a larger number of clinical samples, and possibly from multicentre studies, are needed to further confirm the assays sensitivity, specificity, and reliability. Second, our results cannot be generalizable to the full pathway of SARS-CoV-2 sgRNAs because sgRNA of other genes like S or orf7a were not tested [27]. Another important limitation is that our results did not include samples by Omicron wave, that could be

characterized by different sgRNA kinetics. This limitation avoided the possibility to confirm the performance of the ddPCR assays here described in the Omicron Clade, even if the designed assays target highly conserved SARS-CoV-2 regions, not affected by Omicron variability.

## Conclusion

Overall, we have provided exploratory results demonstrating that sgRNA quantification with a molecular tool characterized by high sensitivity and accuracy could be helpful to estimate the replicative activity of SARS-CoV-2, and can represent a correlate of active infection. The inclusion of this assay in available SARS-CoV-2 diagnostics procedure might help in optimizing patients monitoring and management, at both community and hospital levels. Indeed, application of sgRNA quantification might help in the management of most fragile settings, like immunocompromised patients, known to be at risk for prolonged infection, and persistent SARS-CoV-2 RNA positivity.

## Supporting information

**S1 Fig. Selection criteria for the 51 nasopharyngeal swabs included in the study.**
(PDF)

**S2 Fig. Quantasoft panel for RNAse P housekeeping gene of the 51 nasopharyngeal swabs (one well per patient).** In each panel, the first and second wells represent the negative and positive reaction control, respectively.
(PDF)

**S1 Table. Primers and probes used to quantify SARS-CoV-2 load and subgenomic RNA.**
(DOCX)

**S2 Table. SARS-CoV-2 genomic and subgenomic RNA load against culture isolation.**
(DOCX)

**S3 Table. Demographic and clinical characteristics of patients against SARS-CoV-2 culture isolation vs subgenomic RNA concordance.**
(DOCX)

**S1 File. Supplementary information reporting the full protocol used for the detection and quantification of SARS-CoV-2 subgenomic RNAs and viral load.**
(DOCX)

**S2 File. Supplementary information reporting data related to Figs 1, 2 and S2 Fig.**
(XLSX)

## Acknowledgments

We thank all the staff of the Microbiology and Virology Laboratory of ASST Grande Ospedale Metropolitano Niguarda and of the Microbiology and Virology Department, Fondazione IRCCS Policlinico San Matteo for outstanding technical support in processing swab samples, performing laboratory analyses and data management.

## Author Contributions

**Conceptualization:** Antonio Piralla, Fausto Baldanti, Carlo Federico Perno, Claudia Alteri.

**Data curation:** Rossana Scutari, Silvia Renica, Claudia Alteri.

**Formal analysis:** Rossana Scutari, Silvia Renica.

**Investigation:** Valeria Cento, Arianna Pani, Marco Merli, Diana Fanti, Chiara Vismara, Massimo Puoti, Alessandra Bandera.

**Methodology:** Rossana Scutari, Silvia Renica, Alice Nava, Josè Camilla Sammartino, Alessandro Ferrari, Antonio Piralla.

**Supervision:** Francesco Scaglione, Alessandra Bandera, Andrea Gori, Fausto Baldanti, Carlo Federico Perno, Claudia Alteri.

**Writing – original draft:** Rossana Scutari, Silvia Renica.

**Writing – review & editing:** Andrea Gori, Antonio Piralla, Fausto Baldanti, Carlo Federico Perno, Claudia Alteri.

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
