## [Decision Letter · Decision Letter 0]

6 Mar 2023

PONE-D-22-24987Quantitative SARS-CoV-2 subgenomic RNA as a surrogate marker for viral infectivity: comparison between culture isolation and direct sgRNA quantificationPLOS ONE

Dear Dr. Alteri,

Thank you for submitting your manuscript to PLOS ONE. After careful consideration, we feel that it has merit but does not fully meet PLOS ONE’s publication criteria as it currently stands. Therefore, we invite you to submit a revised version of the manuscript that addresses the points raised during the review process.

We look forward to receiving your revised manuscript.

Kind regards,

Ahmed S. Abdel-Moneim, Ph.D.

Academic Editor

PLOS ONE

5. PLOS requires an ORCID iD for the corresponding author in Editorial Manager on papers submitted after December 6th, 2016. Please ensure that you have an ORCID iD and that it is validated in Editorial Manager. To do this, go to ‘Update my Information’ (in the upper left-hand corner of the main menu), and click on the Fetch/Validate link next to the ORCID field. This will take you to the ORCID site and allow you to create a new iD or authenticate a pre-existing iD in Editorial Manager. Please see the following video for instructions on linking an ORCID iD to your Editorial Manager account: https://www.youtube.com/watch?v=_xcclfuvtxQ.

Reviewers' comments:

Reviewer's Responses to Questions

**Comments to the Author**

1. Is the manuscript technically sound, and do the data support the conclusions?

Reviewer #1: Partly

Reviewer #2: Partly

Reviewer #3: Yes

2. Has the statistical analysis been performed appropriately and rigorously? 

Reviewer #1: No

Reviewer #2: Yes

Reviewer #3: Yes

3. Have the authors made all data underlying the findings in their manuscript fully available?

Reviewer #1: Yes

Reviewer #2: No

Reviewer #3: Yes

4. Is the manuscript presented in an intelligible fashion and written in standard English?

Reviewer #1: Yes

Reviewer #2: Yes

Reviewer #3: Yes

5. Review Comments to the Author

Reviewer #1: The duration of time since symptom onset is extraordinarily long in this study. Usually, it is not possible to culture replicable virus after 10 days since an initial positive test; the median time here is 19 days, and the longest duration was 51 days. The median duration of time since symptom onset was 37 days in the subgroup of patients with clinical information. It is extremely unusual, if not unheard of, to isolate replicable virus this long after symptom onset, even if some of the patients were immunocompromised (not a majority). This makes me wonder if something is wrong with the data or the description of the data, or if I am simply not understanding something.

Other specific comments follow:

--The sensitivity, specificity, NPV, and PPV estimates require confidence intervals-- particularly since they are the primary outcome measures of this study.

--The authors should address more fully the criticism that subgenomic RNAs persist in samples for a long period of time and are thus not suitable as a proxy for infectiousness. The durations since symptom onset presented in this study, however, appear to be much longer than those previously reported (see above).

--The ROC results should be presented before the main results, since the cutoffs obtained in the ROC results are used for all subsequent comparisons.

--It is an acknowledged but real limitation that this study was conducted prior to the emergence of the omicron variant, as subgenomic RNA .

--As noted by the authors, the sample size is quite small, with only 18 culture-positive patients. This renders the results of this study exploratory at best.

--I'm afraid I cannot recommend publication of this article in PLOS ONE owing to the questions I have about the long time between symptom onset and culture positivity, the lack of appropriate measures of uncertainty for sensitivity and specificity calculations, and the the small sample size.

Reviewer #2: The study presents the results of original research. Authors use an original methodical approach for measuring of SARS-CoV-2 sgRNA levels.

But there are some issues with the study design. The main problem is sample selection criteria. Authors mention that they collect nasopharyngeal swabs from SARS-CoV-2 positive patients on 37 day after symptom onset (median, IQR 18-67). Several studies (for example doi: 10.1186/s12916-020-01810-8 and doi: 10.1016/S2213-2600(22)00226-0) have shown that viral shedding in upper respiratory tract usually stops after 14 days post-symptom onset. So, obviously many of swabs (53/110) were PCR-negative and others (39/110) had low viral loads undetectable by virus isolation.

Authors don’t provide data of RNAse-P assay in genomic SARS-CoV-2 RNA-targeted RT-PCR which should act as the Internal Control and this absence doesn’t allow to judge about the quality of sample collection and RNA purification steps.

Also some typos were found: “In detail, genomic RNA assay targeted three different RNA-dependent RNA-polymerase (RdRp) of SARS-CoV-2 and the RNAseP housekeeping gene” (lines 103-105). Do the authors use three PCR assays targeted viral RdRp or do they use one which consists of three oligos targeted RdRp?

The reviewer would recommend to include in the study samples collected from patients within 0-14 days post-symptom onset with different viral loads and exclude samples which are negative on SARS-CoV-2 genome RT-PCR.

Reviewer #3: The manuscript presents convincing data on the possibility of application of sgRNA quantification to determine active viral replication of SARS-CoV-2. The authors present results on the correlation between viral load of genomic and subgenomic RNAs as well as with the culture isolation test and the clinical infection pattern of the patients.

Despite the conclusive data obtained, the authors are correct in pointing out the limitations of the presented work related to the small sample of patients and the inclusion of other genes sgRNA (particularly interesting is the S-protein gene). The authors also point out the possibility of applying this test to people with long-term positivity to SARS-CoV-2, although in this study nasopharyngeal swabs were taken from people with symptoms of the disease (acute infection).

Despite its simplicity, the present work makes a good impression.

A few questions and comments to help improve the manuscript.

1. Was the selected patient tested for the antigen (nucleocapsid protein) or the standard PCR test used in routine clinical practice? This data can be added to the manuscript (e.g., Table 1).

2 By analyzing the data in Table 1, the authors conclude that there is no difference between patients with SARS-CoV-2 positive and negative culture isolation. However, the last column (P-value) with such a random small sample of patients, in my opinion, is unnecessary.

3. The authors do not provide primers for genomic RNA and sgRNA PCR in the "Material and Methods" section. I recommend inserting them as a table in this section, because otherwise a number of questions arise. For example, is the cgRNA sequence part of the complete genomic RNA or not?

4. Since this work is to study the correlation between several parameters, I recommend presenting a graphical design of the study to make it easier to understand the progress of the work.

5. In the "Discussion" section, the results of other researchers on SARS-CoV-2 subgenomic RNA detection and their possible role as a surrogate-to marker of infectious viral shedding could be described in more detail.

6. PLOS authors have the option to publish the peer review history of their article (what does this mean?). If published, this will include your full peer review and any attached files.

---

## [Author Response · Author response to Decision Letter 0]

25 Apr 2023

PONE-D-22-24987

Title: Quantitative SARS-CoV-2 subgenomic RNA as a surrogate marker for viral infectivity: comparison between culture isolation and direct sgRNA quantification

Journal: PLOS ONE

We would like to thank the reviewers for their time and for the constructive criticisms they arose. 

The revisions of the manuscript in accordance with their comments are summarized in our responses below.

Response to Reviewers’ Comments

Reviewer #1: 

1. The duration of time since symptom onset is extraordinarily long in this study. Usually, it is not possible to culture replicable virus after 10 days since an initial positive test; the median time here is 19 days, and the longest duration was 51 days. The median duration of time since symptom onset was 37 days in the subgroup of patients with clinical information. It is extremely unusual, if not unheard of, to isolate replicable virus this long after symptom onset, even if some of the patients were immunocompromised (not a majority). This makes me wonder if something is wrong with the data or the description of the data, or if I am simply not understanding something.

Answer: According with this comment and comments 1 and 4 of the in Reviewer#2, the revised version of the manuscript has been focused on samples belonging to patients with a time between symptom onset and sampling date equal or inferior to 14 days. Therefore, the new analyses included 51 patients (Supplementary Figure 1 and Page 3-4, Lines 85-90 of the main manuscript), characterized by a median time between symptom onset and swab sampling of 8 (1-11) days. 

2. Other specific comments follow:

The sensitivity, specificity, NPV, and PPV estimates require confidence intervals-- particularly since they are the primary outcome measures of this study.

Answer: According with the reviewer’s comment we have included the confidence intervals of all parameters used to define assays accuracy (Page 6, Lines 168-178; Page 7, Lines 202-217). 

3. The authors should address more fully the criticism that subgenomic RNAs persist in samples for a long period of time and are thus not suitable as a proxy for infectiousness. The durations since symptom onset presented in this study, however, appear to be much longer than those previously reported (see above).

Answer: According with the reviewer’s comment a deeper description of the criticism regarding the role of subgenomic RNAs in the proxy for infectiousness has been included in the new version of the manuscript (Page 9, Lines 238-244). As reported in point n. 1 of this revision, samples belonging to patients with a time between symptom onset and sampling date >14 days were excluded from the revised version of the manuscript. The analyses were extensively revised.

4. The ROC results should be presented before the main results, since the cutoffs obtained in the ROC results are used for all subsequent comparisons.

Answer: In our revised manuscript the ROC analysis was performed in order to define the best cut-off of sgRNA load to predict viable virus. The cut-offs estimated by the ROC analysis were then used to perform a speculative comparison with clinical presentation. In line with these considerations the paragraphs “SARS-CoV-2 sgRNA load cut-off definition for viable virus” and “Correlation with clinical presentation” are reported after the description of patients’ characteristics and after the description of SARS-CoV-2 subgenomic results against culture (Page 7, Lines 200-205).

5. It is an acknowledged but real limitation that this study was conducted prior to the emergence of the omicron variant, as subgenomic RNA.

Answer: We are aware that the absence of the Omicron variant in the study is a major limitation. In agreement with the reviewer's comment, we have stressed this limitation in the discussion of the new version of the manuscript (Page 9, Lines 276-279). 

6. As noted by the authors, the sample size is quite small, with only 18 culture-positive patients. This renders the results of this study exploratory at best.

Answer: We agree with the reviewer that the sample size is rather small. We have stressed this limitation in the new version of the manuscript, acknowledging the exploratory nature of the results obtained (Page 9-10; Lines 270-271, 281-183).

7. I'm afraid I cannot recommend publication of this article in PLOS ONE owing to the questions I have about the long time between symptom onset and culture positivity, the lack of appropriate measures of uncertainty for sensitivity and specificity calculations, and the small sample size.

Answer: We would like to thank the reviewer for her/his time and the constructive criticisms she/he has given. All the comments have been carefully considered and addressed. We hope that in this revised version the manuscript can be considered again for potential publication in Plos One. 

Reviewer #2:

The study presents the results of original research. Authors use an original methodical approach for measuring of SARS-CoV-2 sgRNA levels.

But there are some issues with the study design.

Answer: We thank the Reviewer for the appreciation of our work. We revised the study population and the whole manuscript to further improve quality and clarity.

1. The main problem is sample selection criteria. Authors mention that they collect nasopharyngeal swabs from SARS-CoV-2 positive patients on 37 day after symptom onset (median, IQR 18-67). Several studies (for example doi: 10.1186/s12916-020-01810-8 and doi: 10.1016/S2213-2600(22)00226-0) have shown that viral shedding in upper respiratory tract usually stops after 14 days post-symptom onset. So, obviously many of swabs (53/110) were PCR-negative and others (39/110) had low viral loads undetectable by virus isolation.

Answer: In accordance with this comment and comment 1 of the reviewer #1, samples belonging to patients with a time between symptom onset and sampling date longer than 14 days were excluded from the revised version of the manuscript. Therefore, the new analyses included 51 patients (Supplementary Figure 1 and Page 3-4, Lines 85-90 of the main manuscript). The new results showed a median time between symptom onset and swab sampling of 8 (1-11) days. 

2. Authors don’t provide data of RNAse-P assay in genomic SARS-CoV-2 RNA-targeted RT-PCR which should act as the Internal Control and this absence doesn’t allow to judge about the quality of sample collection and RNA purification steps.

Answer: According with the reviewer’s comment, in the new version of the manuscript, we added the Supplementary Figure 2 reporting the Quantasoft panel for RNAse P housekeeping gene of the 51 nasopharyngeal swabs. In addition, in the uploaded supplementary material, we included the individual RNAse-P values of the 51 samples considered in the study. As stated in the manuscript (Page 6, Lines 179-180) RNAseP RNA quantification confirmed the high quality of all the 51 nasopharyngeal swabs.

3. Also some typos were found: “In detail, genomic RNA assay targeted three different RNA-dependent RNA-polymerase (RdRp) of SARS-CoV-2 and the RNAseP housekeeping gene” (lines 103-105). Do the authors use three PCR assays targeted viral RdRp or do they use one which consists of three oligos targeted RdRp?

Answer: In accordance with this comment and reviewer comment 3 of reviewer #3, we clarified this part of the manuscript (Page 4, Lines 106-114) and included a Supplementary Table 3 reporting the sequences of primers and probes used to quantify genomic RNA and sgRNA. In detail, one home-made and two previously tested assays (WHO. Real-Time RT-PCR Assays for the Detection of SARS-CoV-2;) targeting 3 different regions of RNA-dependent RNA-polymerase (RdRp) of SARS-CoV-2 were used to quantify SARS-CoV-2 genomic RNA. The assay targeting the RNAseP housekeeping gene was used as reference (CDC. CDC’s Influenza SARS-CoV-2 Multiplex Assay). The sgRNAs were quantified using assays adapted for the ddPCR system and targeting the envelope and nucleocapsid transcripts (Wölfel R. et al., 2020; Telwattw S. et al., 2022). 

4. The reviewer would recommend to include in the study samples collected from patients within 0-14 days post-symptom onset with different viral loads and exclude samples which are negative on SARS-CoV-2 genome RT-PCR.

Answer: Thanks for this comment. In accordance with comment n. 1 of this revision and comment 1 of the reviewer #1, samples belonging to patients with a time between symptom onset and sampling date >14 days were excluded from the revised version of the manuscript.

Reviewer #3: 

The manuscript presents convincing data on the possibility of application of sgRNA quantification to determine active viral replication of SARS-CoV-2. The authors present results on the correlation between viral load of genomic and subgenomic RNAs as well as with the culture isolation test and the clinical infection pattern of the patients.

Despite the conclusive data obtained, the authors are correct in pointing out the limitations of the presented work related to the small sample of patients and the inclusion of other genes sgRNA (particularly interesting is the S-protein gene). The authors also point out the possibility of applying this test to people with long-term positivity to SARS-CoV-2, although in this study nasopharyngeal swabs were taken from people with symptoms of the disease (acute infection).

Despite its simplicity, the present work makes a good impression.

A few questions and comments to help improve the manuscript.

Answer: We thank the reviewer for his/her positive and thoughtful comments.

1. Was the selected patient tested for the antigen (nucleocapsid protein) or the standard PCR test used in routine clinical practice? This data can be added to the manuscript (e.g., Table 1).

Answer: The patients selected for the study were not tested for antigen (nucleocapsid protein) but for presence of genomic RNA with the standard real-time PCR test used in routine clinical practice. In this regard, it was possible to retrieve information on positivity to the gene targets tested but not the individual CT values. The positivity information shown by the standard PCR at only one target or at multiple targets has been included in the new version of Table 1.

2. By analyzing the data in Table 1, the authors conclude that there is no difference between patients with SARS-CoV-2 positive and negative culture isolation. However, the last column (P-value) with such a random small sample of patients, in my opinion, is unnecessary.

Answer: According with the reviewer’s comment, in the new version of Table 1 we removed the p-values reported. Due to the small sample size, P-values in the Supplementary Table 3 and in the “Correlation with clinical presentation” paragraph were also removed.

3. The authors do not provide primers for genomic RNA and sgRNA PCR in the "Material and Methods" section. I recommend inserting them as a table in this section, because otherwise a number of questions arise. For example, is the cgRNA sequence part of the complete genomic RNA or not?

Answer: In accordance with this comment and comment 3 of reviewer #2, we clarified this part of the manuscript (Page 4, Lines 106-114) and included a Supplementary Table 3 reporting the sequences of primers and probes used to quantify genomic RNA and sgRNA. In detail, one home-made and two previously tested assays (WHO. Real-Time RT-PCR Assays for the Detection of SARS-CoV-2;) targeting 3 different regions of RNA-dependent RNA-polymerase (RdRp) of SARS-CoV-2 were used to quantify SARS-CoV-2 genomic RNA. The assay targeting the RNAseP housekeeping gene was used as reference (CDC. CDC’s Influenza SARS-CoV-2 Multiplex Assay). The sgRNAs were quantified using assays adapted for the ddPCR system and targeting the envelope and nucleocapsid transcripts (Wölfel R. et al., 2020; Telwattw S. et al., 2022). 

4. Since this work is to study the correlation between several parameters, I recommend presenting a graphical design of the study to make it easier to understand the progress of the work.

Answer: In light of the comments of reviewers # 1 and #2, the study design was modified by including only samples belonging to patients with a time between symptoms and sampling collection ≤14 days. Based on these comments and the new selection criteria, a Supplementary Figure 1 describing the flowchart of the study population was added.

5. In the "Discussion" section, the results of other researchers on SARS-CoV-2 subgenomic RNA detection and their possible role as a surrogate-to marker of infectious viral shedding could be described in more detail.

Answer: According with this reviewer comment and point 3 of reviewer 1, a deeper discussion of the role of subgenomic RNAs in the proxy for infectiousness has been included in the new version of the manuscript (Page 9, Lines 257-268).

---

## [Editor Report · Decision Letter 1]

13 Jun 2023

PONE-D-22-24987R1Quantitative SARS-CoV-2 subgenomic RNA as a surrogate marker for viral infectivity: comparison between culture isolation and direct sgRNA quantificationPLOS ONE

Dear Dr. Alteri,

Thank you for submitting your manuscript to PLOS ONE. After careful consideration, we feel that it has merit but does not fully meet PLOS ONE’s publication criteria as it currently stands. Therefore, we invite you to submit a revised version of the manuscript that addresses the points raised during the review process.

We look forward to receiving your revised manuscript.

Kind regards,

Ahmed S. Abdel-Moneim, Ph.D.

Academic Editor

PLOS ONE

---

## [Author Response · Author response to Decision Letter 1]

12 Jul 2023

PONE-D-22-24987R1

Title: Quantitative SARS-CoV-2 subgenomic RNA as a surrogate marker for viral infectivity: comparison between culture isolation and direct sgRNA quantification

Journal: PLOS ONE

We would like to thank academic editor for his time and for the constructive criticisms he arose. 

The revisions of the manuscript in accordance with his comment are summarized in our responses below.

Response to Editor’ Comment

1. We recommend that you deposit your laboratory protocols in protocols.io to enhance the reproducibility of your results. Protocols.io assigns your protocol its own identifier (DOI) so that it can be cited independently in the future. For instructions see: https://journals.plos.org/plosone/s/submission-guidelines#loc-laboratory-protocols. Additionally, PLOS ONE offers an option for publishing peer-reviewed Lab Protocol articles, which describe protocols hosted on protocols.io. Read more information on sharing protocols at https://plos.org/protocols?utm_medium=editorialemail&utm_source=authorletters&utm_campaign=protocols.

Answer: According with the editor’s comment, we provided the full protocol describing the detailed methodology used for the detection and quantification of SARS-CoV-2 subgenomic RNAs and viral load in the revised version of the manuscript as Supplementary File 1 (Page 4, Lines 114-115). We tried to deposit the protocol in protocols.io as the editor suggested, but we did not received the customer code required for the submission and requested to plosone@plos.org by email in June, 2023, following the lab protocol guidelines (available at https://journals.plos.org/plosone/s/submission-guidelines#loc-lab-protocols). We also added a funding and data availability statement to the revised manuscript (Page 11, lines 323-329).

---

## [Decision Letter · Decision Letter 2]

23 Aug 2023

Quantitative SARS-CoV-2 subgenomic RNA as a surrogate marker for viral infectivity: comparison between culture isolation and direct sgRNA quantification

PONE-D-22-24987R2

Dear Dr. Alteri,

We’re pleased to inform you that your manuscript has been judged scientifically suitable for publication and will be formally accepted for publication once it meets all outstanding technical requirements.

Kind regards,

Ahmed S. Abdel-Moneim, Ph.D.

Academic Editor

PLOS ONE

Additional Editor Comments (optional):

Reviewers' comments:

Reviewer's Responses to Questions

**Comments to the Author**

1. If the authors have adequately addressed your comments raised in a previous round of review and you feel that this manuscript is now acceptable for publication, you may indicate that here to bypass the “Comments to the Author” section, enter your conflict of interest statement in the “Confidential to Editor” section, and submit your "Accept" recommendation.

Reviewer #4: All comments have been addressed

2. Is the manuscript technically sound, and do the data support the conclusions?

Reviewer #4: Yes

3. Has the statistical analysis been performed appropriately and rigorously? 

Reviewer #4: Yes

4. Have the authors made all data underlying the findings in their manuscript fully available?

Reviewer #4: Yes

5. Is the manuscript presented in an intelligible fashion and written in standard English?

Reviewer #4: Yes

6. Review Comments to the Author

Reviewer #4: The text of the manuscript was modified according to the reviewers' comments. This revision version of the MS may be published in the journal PlosOne.

7. PLOS authors have the option to publish the peer review history of their article (what does this mean?). If published, this will include your full peer review and any attached files.

Reviewer #4: No

---

## [Editor Report · Acceptance letter]

25 Aug 2023

PONE-D-22-24987R2 

Quantitative SARS-CoV-2 subgenomic RNA as a surrogate marker for viral infectivity: comparison between culture isolation and direct sgRNA quantification 

Dear Dr. Alteri:

I'm pleased to inform you that your manuscript has been deemed suitable for publication in PLOS ONE. Congratulations! Your manuscript is now with our production department. 

Kind regards, 

on behalf of

Prof. Ahmed S. Abdel-Moneim 

Academic Editor

PLOS ONE